# Novel Therapeutic Devices in Heart Failure

**DOI:** 10.3390/jcm11154303

**Published:** 2022-07-25

**Authors:** Mateusz Guzik, Szymon Urban, Gracjan Iwanek, Jan Biegus, Piotr Ponikowski, Robert Zymliński

**Affiliations:** Institute of Heart Diseases, Wroclaw Medical University, 50-556 Wroclaw, Poland; mateuszguzik23@gmail.com (M.G.); giwanek95@gmail.com (G.I.); janbiegus@gmail.com (J.B.); piotr.ponikowski@umw.edu.pl (P.P.); robertzymlinski@gmail.com (R.Z.)

**Keywords:** heart failure, cardiorenal syndrome, autonomic dysregulation, respiratory disturbances, novel devices

## Abstract

Heart failure (HF) constitutes a significant clinical problem and is associated with a sizeable burden for the healthcare system. Numerous novel techniques, including device interventions, are investigated to improve clinical outcome. A review of the most notable currently studied devices targeting pathophysiological processes in HF was performed. Interventions regarding autonomic nervous system imbalance, i.e., baroreflex activation therapy; vagus, splanchnic and cardiopulmonary nerves modulation; respiratory disturbances, i.e., phrenic nerve stimulation and synchronized diaphragmatic therapy; decongestion management, i.e., the Reprieve system, transcatheter renal venous decongestion system, Doraya, preCardia, WhiteSwell and Aquapass, are presented. Each segment is divided into subsections: potential pathophysiological target, existing evidence and weaknesses or unexplained issues. Novel therapeutic devices represent great potential in HF therapy management; however, further evidence is necessary to fully evaluate their utility.

## 1. Introduction

Heart failure (HF) is a clinical syndrome resulting from structural and/or functional abnormality of the heart, leading to elevated intracardiac pressures and/or insufficient cardiac output. Increased cardiac filling pressures and neuro-hormonal disturbances resulting in fluid retention and redistribution are major factors responsible for congestion development and acute decompensation in heart failure [1].

As the HF pathophysiology is multidimensional, device interventions allow direct or indirect targeting of biological HF pathways, e.g. methods to manipulate sympathetic nervous system (SNS) imbalance, respiratory dysregulation or volume overload have been developed (Table 1). To preserve the article’s coherence and compactness, we decided not to describe all promising techniques, but we focused on selected pathophysiological processes crucial in HF (Figure 1).

HF remains a major medical problem and is associated with a high occurrence of rehospitalization and deaths, which constitute a huge problem for patients as well as healthcare systems worldwide [19]. Given that, numerous methods to improve outcome in HF have arisen, some including device-based treatment techniques.

Novel devices are supported by a strong theoretical background and a number of positive early signs from several small studies. Nevertheless, all device therapies, especially those that are permanently implanted in the patient, should undergo thorough assessment in large-scale prospective studies before they can be used in clinical practice.

## 2. Targeting Autonomic Nervous System Regulation

### 2.1. Potential Pathophysiological Target

Physiologically, the autonomic nervous system (ANS) may be described as a highly dynamic structure, driven by uncountable neurohormonal reactions to maintain homeostasis. The imbalance of the ANS plays a crucial role in the pathogenesis of HF as the SNS exceeds the buffer capabilities of the parasympathetic nervous system (PNS). The ANS is responsible for modulation of the heart rate, systemic vascular resistance, arterial blood pressure and cardiac afterload, whereby constant overactivity of SNS leads to undesired maladaptations and cardiovascular remodeling. This phenomenon is reflected in the treatment of HF. From the clinical point of view, there are several possible targets for ANS modulation. Modulation of selected subtypes of receptors (e.g., baroreflex activation therapy) allows for interaction with specific ANS branches (sympathetic or parasympathetic). Via the afferent nerves, stimuli are transmitted from receptors to the central nervous system (CNS). On this level, impulses are analyzed and transferred to the effector pathways. The efferent nerves transmit impulses from the CNS to the neurochemical synapses. Modulation of this process directly influences PNS (Vagus nerve stimulation) or SNS (Splanchnic nerve modulation). In the end, impulses reach the presynaptic membrane resulting in the secretion of neurochemical transmitters (e.g., epinephrine, norepinephrine and acetylcholine), which react with receptors localized in the effector tissue. Crucial for HF is the overactivity of SNS mediated by adrenergic receptors [20]. Numerous studies of beta-adrenergic receptor blockers have proven their impact on survival in HFpEF patients [21,22]. Additionally, the SNS is directly connected with the Renin-Angiotensin-Aldosterone system (RAAS), responsible for increased sodium and water reabsorption with subsequent fluid accumulation, which elevates cardiac filling pressure and promotes congestion development, the indisputable targets of HF therapy [1]. Although the role of the SNS in HF is certain, the knowledge about its mechanisms responsible for HF is still unclear, and the ANS is an area for ongoing research in HF therapies especially using novel biomedical technologies.

### 2.2. Baroreflex Activation Therapy

Baroreflex activation therapy (BAT) uses a physiological reflex pathway to rebalance the activity of the ANS. Electrical stimulation of the carotid bodies sends afferent nerve impulses to the CNS that reacts by increasing PNS firing and decreasing SNS outflow [23]. The cardiovascular system response is acute and results in the decrease of heart rate and systemic vascular resistance with subsequent reduction in both systolic and diastolic blood pressure [23].

#### 2.2.1. Existing Evidence

Several clinical studies have evaluated the effectiveness and safety of BAT. A multicenter, prospective, randomized, controlled trial–Baroreflex Activation Therapy for Heart Failure (BeAT-HF, NCT02627196)–showed that in the group of 264 patients with the FDA-approved enrolment criteria for BAT (EF ≤ 35%, NT-proBNP < 1.600 pg/mL, NYHA functional class III and without Class I indication for CRT), BAT is a safe procedure that significantly improves quality of life, exercise capacity and functional status, while it decreases NT-proBNP and reduces the number of HF hospitalizations per year. The study reported that the overall major adverse neurological and cardiovascular event-free rate was 97.2%, while the system and procedure-related complication event-free rate was 85.9% [2]. Cardiovascular mortality and HF morbidity rates are still under investigation (1200 participants, 5 years of observation, NCT02627196) Dell’Oro et al. demonstrated that in the group of seven patients who completed follow-up, BAT significantly improved EF (from 32.3 ± 2 to 36.7 ± 3% in 43 months, *p* < 0.05) and reduced heart failure-related hospitalization rate. There were no side effects reported in this study [3]. Apart from HF, BAT is also widely investigated as a potential drug-resistant arterial hypertension treatment [23].

#### 2.2.2. Weaknesses or Unexplained Issues

Despite positive early results, there is a need for further, well-powered clinical trials before BAT can be incorporated into HF clinical practice. BAT needs at least larger-scale research that includes longer follow-up, a higher number of patients and clarified outcomes with mortality risks [24]. The study performed by Dell’Oro et al. was not registered as a clinical trial.

### 2.3. Vagus Nerve Stimulation

Vagus nerve stimulation (VNS) is an autonomic system modulation that aims to level autonomic system imbalance by increasing PNS activity. Electrostimulation of the easily accessed right cervical vagus nerve induces neurohormonal reactions that buffer the overactivity of SNS [25].

#### 2.3.1. Existing Evidence

The Neural Cardiac Therapy for Heart Failure (NECTAR-HF, NCT01385176, 95 participants, 63 randomized to therapy) trial was the first study that evaluated the usefulness of VNS in HFrEF. It showed improvements in quality of life, NYHA class and exercise capacity without changes in echocardiographic measures (primary endpoint defined as the change in left ventricle end-systolic diameter) in the VNS treated patients. There were no significant differences in the serious adverse event (SAE) rates between the control and therapy groups. The overall rate of implantation-related infections was 7.4% [4]. The Autonomic Regulation Therapy for the Improvement of Left Ventricular Function and Heart Failure Symptoms (ANTHEM-HF, NCT01823887, 60 participants) uncontrolled design study delivered information about the safety of this procedure, and it showed positive, durable improvements in cardiac function and echocardiography parameters after 6 months of treatment. Additionally, this study confirmed significant improvement in NYHA functional class and exercise tolerance. One death related to the device implantation procedure caused by an embolic stroke that occurred 3 days after surgery in a patient suffering from extensive atherosclerosis of the carotid arteries was reported [5]. The promising application of VNS may be heart rate-dependent stimulation, which, apart from balancing the autonomic system, restores physiological relations [26].

#### 2.3.2. Weakness or Unexplained Issues

Although VNS has a significant positive impact on a patient’s functional status, it does not impact the prognosis [27]. The ANTHEM-HF study was conducted without a control group, which is a significant limitation. To exclude the placebo effect and assess the safety of the procedure, there is a need for a randomized, controlled clinical trial [5]. Moreover, positive echocardiographic changes are not reported by any studies [27]. Interestingly, positive functional changes observed during VNS therapy are not accompanied by NT-proBNP serum level decrease.

### 2.4. Splanchnic Nerve Modulation

The splanchnic nerves are responsible for autonomic innervation of the upper abdominal viscera (e.g., liver) and are highly connected with splanchnic vascular volume management, primarily caused by visceral vasoconstriction during exercise. The visceral vascular bed is a natural reservoir of blood volume that can be quickly relocated for an urgent need (like hypovolemia, hemorrhage, or exercise). Redistribution of blood volume from the extra-thoracic compartments into the central circulation is believed to be a significant contributor to elevated filling pressures in HF patients, including HF with preserved ejection fraction (HFpEF) [8]. Modulation (blockage or partial blockage) of the splanchnic nerves (SNM) decreases sympathetic tone. It thereby prevents the rapid shift of blood from the splanchnic bed to the central circulation during physical exercise.

SNM may protect the central venous system from acute volume redistribution and subsequent cardiac filling pressure increase [28]. SNM is reached by uni- or bilateral chemical, electrical or surgical greater splanchnic nerve blockage.

#### 2.4.1. Existing Evidence

The splanchnic-HF 1 (NCT02669407) and 2 (NCT03453151) trials reported promising effects of SNM therapy in both acute decompensated (ADHF) and chronic heart failure (CHF). Eleven ADHF patients with advanced HFrEF underwent bilateral temporary percutaneous splanchnic nerve block with lidocaine. In this group, significant reduction in pulmonary capillary wedge pressure (from 30  ±  7 mmHg at baseline to 22  ±  7 mmHg at 30 min, *p*  <  0.001) and an increase in cardiac index (from 2.17 ± 0.74 L/min/m^2^ at baseline to 2.59 ± 0.65 L/min/m^2^ at 30 min *p*  =  0.007) were reported [6]. Similar findings were provided by a study of 18 CHF patients who underwent the same procedure [7]. In HFpEF, permanent ablation of the right greater splanchnic nerve resulted in the reduction of intracardiac filling pressures during exercise, as early as 24 h after the procedure [29]. Moreover, a European two-center study investigated the feasibility of permanent surgical right-sided SNM for the treatment of HFpEF (Surgical Resection of the Greater Splanchnic Nerve in Subjects Having Heart Failure with Preserved Ejection Fraction, NCT03715543) demonstrated a significant reduction of PCPW at a 3-month follow-up and significant improvement in NYHA class and quality of life at 12 months after the procedure [28]. The early results of the REBALANCE-HF study (NCT04592445, the ongoing multicenter evaluation of splanchnic ablation for volume management in HFpEF) delivered auspicious results. In the group of 18 enrolled patients, the 20 W exercise PCWP and peak exercise PCWP decreased significantly 1 month after the procedure. At least one NYHA class improvement was experienced by 39% of patients at 1 month and 50% at 3 months after the SNM procedure. This study reported three non-serious device-related adverse events (AE): HF decompensation due to periprocedural fluid overload, transient hypertension and back pain following ablation [8].

#### 2.4.2. Weakness or Unexplained Issues

Safety and efficacy of SNM in the treatment of HF needs to be further investigated. Current scientific reports are based on small patient populations and very limited follow-ups. Notably, the abovementioned studies were proof-of-concept clinical trials without a control group. Additionally, a unified procedure for HF SNM application must be established [28].

### 2.5. Cardiac Pulmonary Nerve Stimulation

This method uses anatomical relations between pulmonary arteries and the cardiac autonomic system elements. An endovascular delivered electrode placed in pulmonary arteries stimulates the surrounding autonomic nerves resulting in positive lusitropic (increasing relaxation of the myocardium during diastole) and positive inotropic (increasing myocardial contractility) effects without an influence on heart rate. Thus, this percutaneous device has at least theoretical potential to improve cardiac function and systemic perfusion and facilitate decongestion in ADHF [9].

#### 2.5.1. Existing Evidence

The first in-human, proof-of-concept, uncontrolled study (NCT04814134) revealed promising cardiac pulmonary nerve stimulation (CPNS) effects. CPNS in HF resulted in LV contractility improvement and an increase in mean arterial pressure without affecting the heart rate. Moreover, the CPNS 2 Feasibility Study demonstrated short-term safety (no SAE reported) and feasibility in chronic HF patients undergoing a catheterization procedure or implantable cardioverter-defibrillator/cardiac resynchronization therapy implantation [9].

#### 2.5.2. Weakness or Unexplained Issues

CPNS is a concept that needs further investigation. Well organized clinical trials are required to provide information about CPNS effectiveness, safety and impact on outcomes.

## 3. Respiratory Disturbances in Heart Failure

The function of the respiratory system is essential not only in the context of the exchange of respiratory gases but also in generating resistance in the pulmonary circulation and pressure changes inside the chest. The constellation of these factors affects the function of the heart itself and the entire circulatory system.

Sleep-disordered breathing is a common pathology, especially in patients with HF, affecting both cardiovascular and respiratory systems. There are two main types of sleep apnea syndromes: obstructive sleep apnea syndrome (OSA) and central sleep apnea syndrome (CSA) [30].

### 3.1. Potential Pathophysiological Target

OSA/CSA increases SNS, RAAS activation, oxidative stress, cell apoptosis, endothelial dysfunction and, as a result, remodeling and fibrosis of the heart [31,32]. These effects are common to the OSA/CSA and HF pathophysiology and accelerate HF progression, despite different mechanisms leading to these consequences [33].

In CSA, the lack of respiration is caused by pathological pauses in neurological impulses triggering breathing muscles contraction, which results in periods of apnea [34,35]. It was found that the underlying cause of this pathology is the augmented ventilation response to the high partial pressure of CO2 (pCO2), also enhanced by hypoxia, especially in acute heart failure (AHF) [33]. Thus, hyperventilation occurs during sleep (as a response to high pCO2), leading to a periodic drop in pCO2, which goes below the threshold for triggering the action potential in the respiratory center. As a result, patients present periodic apnea during night rest [34,35,36].

The OSA is caused by excessive laxity and, as a result, the upper airways collapse during breathing. Several methods of treatment such as continuous positive airway pressure (CPAP), adaptive servo-ventilation, oral inserts, surgical treatment (e.g., uvulopalatopharyngoplasty or maxillomandibular advancement, tracheostomy or hypoglossal nerve stimulation have been proposed [37]. Nevertheless, meta-analyses showed that treatment with CPAP/ASV improved HF patients’ quality of life, with no impact on survival or rehospitalizations. On the other hand, there are signals that the use of ASV in patients with HFrEF and CSA may be even harmful and associated with an increase in all-cause mortality [38,39].

Thus, the main problem in the HF population is the group of patients with CSA, in which there are regular/cyclic pauses in breathing during sleep due to a lack of respiratory effort. Since the act of breathing is mainly caused by the intercostal muscles and the diaphragm, and the cause of the dysfunction lies in the area of the respiratory center, a method of stimulating the phrenic nerve or diaphragm has been proposed for treatment.

### 3.2. Phrenic Nerve Stimulation

Technically, this method is similar to classic cardiac stimulation. An electrode is implanted into a brachiocephalic or pericardiophrenic vein to sense the diaphragm’s contractions during breathing and stimulate the diaphragmatic nerve during apnea. The electrode is connected to the subcutaneously implanted management module.

The task of this device is to maintain a relatively stable pO2 and pCO2 and prevent over-activation of SNS and RAAS. [10,40].

#### 3.2.1. Existing Evidence

The remedē System Pivotal Trial (NCT01816776) was a multicenter, randomized study with 151 participants. It was meant to provide phrenic nerve stimulation and demonstrated a significant reduction in the apnea-hypopnea index (AHI), central apnea index, arousal index, oxygen desaturation ≥4% index, percentage of sleep with rapid eye movement and sleepiness (Epworth Sleepiness Scale (ESS)) [41]. Those findings were sustained in a 5-year follow-up [11].

Costanzo et al. found that patients treated with phrenic nerve stimulation had an improvement in life quality and improvement in left ventricle ejection fraction (LVEF), with no significant difference in end-systolic and end-diastolic volumes [10].

This method was relatively safe. In follow-ups, the AE were most common during the first year and predominantly included electrode dysfunction, electrode dislocation and infection of the implantation site. Cumulatively, in 5-year observations, the SAE occurred in 14% of patients. There was one episode of inadequate intervention by the high-energy implantable device related to hypersensitivity, which was resolved by changing the device settings [10,41,42].

#### 3.2.2. Weaknesses or Unexplained Issues

The effect of phrenic nerve stimulation on mortality in HF patients with CSA syndrome is unknown and large scale clinical trials are required.

### 3.3. Synchronized Diaphragmatic Therapy

Elevation of intrathoracic pressure causes chronic stress on the heart muscle and may worsen HF. The respiratory muscles can significantly influence intrathoracic pressure. Thus, a strategy for synchronic diaphragm stimulation was proposed. It involves implanting a device connected with an electrode that senses the heart rhythm and stimulates the diaphragm.

This system aims to synchronize the cardiac work cycle to changes in diaphragm movement by stimulation of diaphragm’s muscle fibers (especially type I), causing cyclical changes in their tension, which in turn reduces intra-thoracic pressure. It is imperceptible for the patient, as it does not cause contraction of the diaphragm leading to respiratory movement. Thus, it does not cause any discomfort to the patient. In the first study, entitled Epiphrenic II, [12] the electrode-to-diaphragm stimulation was implanted during coronary artery by-pass grafting procedures [12,40]. A minimally invasive method of laparoscopic implantation, which minimizes the risk of complications and shortens the hospitalization period after implantation, has further been developed.

#### 3.3.1. Existing Evidence

In Epiphrenic II (NCT00769678), a randomized study conducted on 33 participants, researchers found improvement in LVEF and HF symptoms on the NYHA scale. There was also an observed increase in maximal power and oxygen consumption during exercise testing. However, no significant improvement in the 6-min walking test (6 MWT) and BNP concentration was recorded in a group with optimized synchronized diaphragmatic stimulation. No SAE were observed [12].

In the VisOne Heart Failure non-randomized study (NCT03484780, 15 participants) improvement of LVEF and quality of life (evaluated in SF-36) and extended walking distance during the 6 MWT were observed at the 1-year follow-up. Best results were achieved in patients with over 80% diaphragm pacing synchronized to the heart cycle. No AE were observed at 12-month follow-up (primary and secondary endpoint) [12,13]. 

#### 3.3.2. Weaknesses or Unexplained Issues

The VisOne study was non-randomized, and both studies were conducted in a small group of patients. Due to the promising results of the trials, it would be worth performing further studies on an extensive study group with a control population.

## 4. Novel Techniques to Facilitate Decongestion

### 4.1. Potential Pathophysiological Target

Loop diuretics remain the cornerstone of the decongestive therapy in HF; however, reduced responsiveness to them, especially in chronic use, constitutes a clinical challenge. Up to nearly 50% of the classically treated HF patients are discharged with residual congestion, which worsens prognosis [43,44]. Extracorporeal ultrafiltration has been proposed as an alternative for pharmacotherapy; however, current results about its safety and the advantage over standard care remain unclear [45]. Given all the exposed deficiencies, interest in novel fluid removal techniques has emerged.

### 4.2. Reprieve Therapy^®^

Reprieve therapy is a method which intends to provide a solution for the more accurately controlled decongestion for HF patients. The Reprieve System is designed to measure the urine output (via urinary catheter) and deliver (adjusted to urine output) a precise volume of replacement solution (via peripheral vein cannula) to achieve the preset fluid balance [14]. This technique is meant to decrease the risk of intravascular volume depletion, which is a strong inner signal for urine output drop during decongestion. The urine output is unpredictable in HF, thus, some patients have large urine outputs that may unintentionally lead to intravascular volume depletion and to so-called diuretic resistance. The Reprieve system is meant to prevent excessive intravascular fluid removal and subsequent volume depletion, which may lead to hypovolemia and hemodynamic instability.

#### 4.2.1. Existing Evidence

TARGET-1 and TARGET-2 studies have assessed the safety and efficacy of controlled decongestion by the Reprieve System in AHF patients (NCT05015764). In both studies, patients in the study group achieved higher urine output, reduction in body weight and a decrease in central venous pressure (CVP), in comparison to the status before the initiation of the therapy. It is noteworthy that, while achieving greater fluid loss, the treatment was safe–systolic blood pressure remained stable. No renal injury makers or a decrease in renal function was observed. There were no SAE, and the most frequent AE was hypokalemia–mean serum potassium dropped from 4.1 to 3.6 mmol/L (*p* < 0.05).

#### 4.2.2. Weaknesses or Unexplained Issues

Data about the Reprieve System comes from two non-randomized, relatively small, prospective single-center studies. Further trials, including randomized controlled trials, are warranted to confirm its value and impact on the outcome, i.e., mortality or HF hospitalizations. Moreover, Reprieve is targeted at AHF patients with preserved diuresis who respond to diuretics. Whether the device holds promise for the facilitation of decongestion in AHF needs further investigation. The new and more advanced device versions are being investigated.

### 4.3. Transcatheter Renal Venous Decongestion System (TRVD) and Doraya Catheter

As renal vein congestion has been assessed as the most critical factor responsible for the worsening renal function in AHF patients [46], attempts to create novel interventions for renal decongestion have arisen. The novel concept of the renal tamponade caused by the congestion, which additionally impedes the renal outflow and subsequently harms renal function, just added importance to the issue [47]. The transcatheter renal venous decongestion system (TRVD) is inserted through a femoral vein catheter-mounted flow pump, the aim of which is to reduce the pressure in the renal veins to the selected target [48]. The device was tested in a porcine model, where renal pressure was artificially increased by a suprarenal balloon and then reduced by the TRVD, showing an increase in renal flow and subsequently an increase in urine output. The trial to evaluate TRVD in the AHF population (NCT03621436) was terminated prematurely due to the sponsors’ decision, and no study results have been published by now.

The Doraya Catheter is deployed in the inferior vena cava below the renal veins. The Doraya catheter was developed to temporarily reduce renal venous pressure by creating a controllable gradient in the inferior vena cava below the renal veins. The device aims to decrease renal venous pressure at the cost of transitory obstruction of the venous outflow from the lower extremities. By partially blocking venous flow, the Doraya creates a gradient of pressure below and above the catheter, which results in a pressure decrease in renal veins and further diminishes the right ventricle preload.

#### 4.3.1. Existing Evidence

The results of the first in-human studies of Doraya are promising (NCT03234647) [15]. No device malfunctions were observed, and all the technical aspects regarding the device deployment and removal were successful. Significant pressure reduction above the catheter was observed as well as a positive diuretic response. Clinical signs of congestion, including dyspnoea, all improved. No device-related or embolic events were observed during the procedure. In a follow-up after 30 days, one SAE was observed, i.e., bleeding hematoma from the injection site, that resolved without sequelae. The Doraya catheter seems to provide an exciting concept for the treatment of AHF patients with inadequate response to the standard diuretic treatment.

#### 4.3.2. Weaknesses or Unexplained Issues

Currently, only pilot studies in a small population, regarding novel renal veins decongestion techniques, have been performed. Studies included a limited population and aimed to assess the feasibility of such strategies, rather than their clinical efficacy and impact on outcome. Further research (which is under way) is necessary to establish the clinical value of the methods mentioned above.

### 4.4. preCARDIA

The producers of the preCARDIA system proposed a distinct approach for congestion relief therapy. The device is inserted into the superior vena cava to cause intermittent occlusion, leading to a decrease in right ventricular preload.

#### 4.4.1. Existing Evidence

The VENUS-HF early feasibility study (NCT03836079) showed a decrease in right atrial pressure and PCWP compared to the pretreatment values. At 24 h of treatment, a 130 and 156% increase in the urine output and net fluid output, consequently, was observed. No device- or procedure-related SAE were observed [16]. Prior studies have also reported its safety in the preclinical model, in terms of thrombotic events, strokes or neurologic deficits. No examined animal has experienced increased cerebral oedema or thrombotic event [49].

#### 4.4.2. Weaknesses or Unexplained Issues

The studies are the first in-human trials of the device. They had a non-randomized design and included a limited number of patients observed for a short period of time. Furthermore, larger studies with prolonged follow-up are warranted to evaluate the safety and precise clinical utility of the preCARDIA system and its impact on outcome.

### 4.5. WhiteSwell^®^

The role of the lymphatic system in HF pathophysiology has been underestimated, but it appears that it could play a role in decongestive therapy. Firstly, lymphatic drainage is essential for interstitial fluid removal. Furthermore, increased central venous pressure disturbs the lymph outflow through the thoracic duct and additionally stimulates lymph production, leading to oedema deterioration [50]. These pathological aspects prompted researchers to create an intervention, which would target the lymphatic flow in HF therapy. WhiteSwell is a device designed to create a low-pressure area in the outflow of the thoracic duct into the venous system. Such a technique aims to facilitate interstitial drainage with simultaneous intravascular fluid removal by diuretic therapy [17].

#### 4.5.1. Existing Evidence

The WhiteSwell (NCT02863796) has been investigated in a sheep model and in one in-human case. In all studied sheep, WhiteSwell was successfully implanted and removed. The desired pressure gradient was achieved. As opposed to the controls with no implanted device, in studied sheep, WhiteSwell not only stopped the further fluid accumulation (understood as the extravascular lung water changes), but effectively initiated its removal [17]. No evidence of hemolysis was noted.

By now, one case of in-human implementation of the device was reported with positive early signals (in terms of serum creatinine, NT-proBNP and CVP change) of the intervention. After the procedure, the patient felt well and reported improvement in the orthopnea and oedema. No AE were reported.

WhiteSwell, and the general perspective for incorporating the lymphatic system into the HF therapy, constitute a promising supplementation to the traditional, intravascular space-based approach.

#### 4.5.2. Weaknesses or Unexplained Issues

Except for all the limitations stemming from the animal model study, some issues need to be solved before wider clinical implementation. No reliable data about the impact of lymphatic system interventions and clinical outcome in HF patients is available. There were also some technical issues regarding the catheter implantation, and the second-generation catheter is now being constructed [17,44].

### 4.6. AquaPass

The AquaPass system has proposed another novel approach for direct interstitial fluid and sodium removal. The Aquapass system enhances sweat rate and thus fluid removal. It is a wearable machine constructed to increase the skin temperature of the lower parts of the body, with no effect on the core temperature [18].

#### Existing Evidence

The AquaPass system was evaluated in a study (NCT04578353) including only 6 healthy subjects and 10 HF patients who underwent three treatment sessions for up to 4 h. The skin temperature increased, with no change in core temperature. The median weight loss was 219 ± 67 g/h, and heart rate, systolic and diastolic pressure remained stable. No AE occurred. Enhancing sweat rate in HF patients seems to be a safe possibility for decongestive therapy; however, further studies are warranted to evaluate the precise value of the method and its impact on outcome [18].

## 5. Limitations

Our study is not free from limitations. Importantly, this is a literature review and was not performed in accordance with systematic review guidelines. Furthermore, to preserve the article compactness, we decided not to include all the promising device-based techniques applied in HF, such as valvular interventions, atrial shunting or cardiac contractility modulation.

## 6. Conclusions

The abovementioned techniques intend to leverage the pathophysiological aspects of heart failure, which have not been used in therapy by now. Notwithstanding the enormous potential of novel approaches, most are still distant from broad clinical appliance. Further, well-designed, randomized, controlled trials are warranted to evaluate their precise value in HF management.

## Figures and Tables

**Figure 1 jcm-11-04303-f001:**
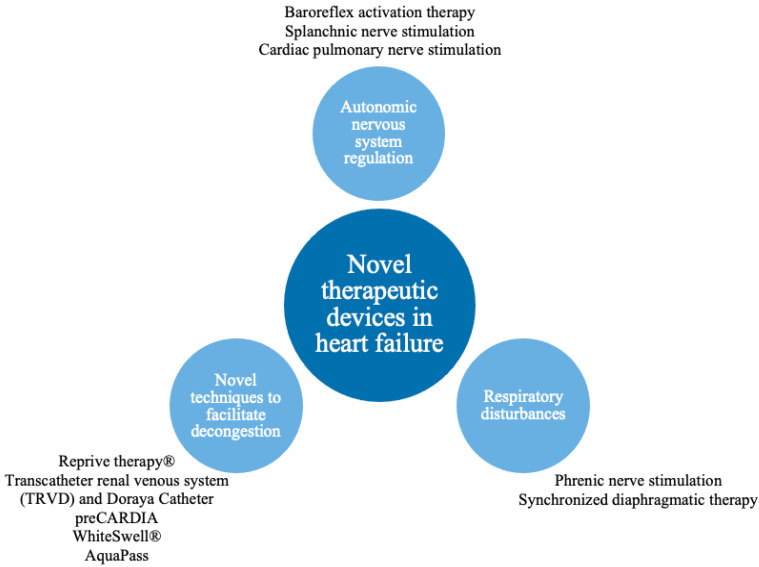
Pathophysiological pathways addressed by novel therapeutic devices.

**Table 1 jcm-11-04303-t001:** Summary of the proposed novel methods.

Method	Pathophysiological Mechanism	Solution	Trial Design and Size	Primary Outcomes	Evidence	Adverse Events
Baroreflex activation therapy	Overactivity of SNS (increased heart rate, arterial pressure, RAAS activity and negative cardiac remodeling).	Stimulation of carotid bodies to restore autonomic system balance.	Multicenter, prospective, controlled trial *n* = 408	Rate of cardiovascular and HF morbidity, MANCE, Change in: NT-proBNP, 6 MHW, MLWHF QOL	BeAT-HF showed improvements of quality of life, exercise capacity, functional status and decrease of NT-proBNP [2]	MANCE event-free rate: 97%. A system or procedure-related serious adverse event occurred in seven patients.
			Single-center, open-label *n* = 11	Not reported	Dell’Oro et al. demonstrated significant improvement of EF and reduction in hospitalization [3]	No adverse effects were reported.
Vagus nerve stimulation	Overactivity of SNS (increased heart rate, arterial pressure, RAAS activity and negative cardiac remodeling).	Increase of PNS activity.	Multicenter, prospective, randomized, controlled trial *n* = 95	Change in LVESD, Percentage of surviving patients.	NECTAR-HF presented significant improvement in quality of life, NYHA class and functional status [4]	There were no significant differences in serious adverse events between control and therapy groups. The overall rate of implantation-related infections was 7.4%
			Multicenter, open-label, uncontrolled trial *n* = 60	Change in: LVESV EF, Adverse events.	ANTHEM-HF showed positive, durable improvement of cardiac function [5]	Serious adverse events occurred in 16 patients. There was one death related to system implantation due to an embolic stroke that occurred 3 days after surgery.
Splanchnic nerve stimulation	Excessive cardiac filling pressure due to overactivity of SNS resulting in visceral vasoconstriction and rapid volume shift from visceral to central compartment during exercise.	GSN modulation preventing exercise provoked visceral vasoconstriction and subsequent fluid shift from the visceral compartment to the central venous system.	Single-center, prospective, open-label, uncontrolled trials *n* = 11, *n* = 15	Change in CVPPAMP PCWP	Splanchnic-HF 1, and Splanchnic-HF 2 showed a reduction in PCPW and improvement of the cardiac index during exercise [6,7]	No adverse events were reported.
			Multicenter, prospective, uncontrolled, pilot study	Change in: mean PCPW at rest and exercise (20 W). Adverse events.	REBALANCE-HF confirmed the reduction in exercise PCPW in HFpEF and NYHA class improvement [8]	There were three non-serious device-related adverse events reported in this study: HF decompensation due to periprocedural fluid overload, transient hypertension and back pain following ablation.
Cardiopulmonary nerve stimulation	Impaired LV contractility and relaxation.	Stimulation of the autonomic system area responsible for LV contractility resulting in positive lusitropic and inotropic effects.	Single-center, first-in-human, proof-of-concept study *n* = 15	Adverse events.	A proof-of-concept study showed improvement of LV contractility and an increase in mean arterial pressure without affecting the heart rate [9]	No device-related serious adverse events were reported.
Phrenic nerve stimulation	Central apnea due to periodic drop in CO_2_ partial pressure to below the threshold for triggering the action potential in the respiratory center caused by greater sensitivity to carbon dioxide leading to potent stimulus of rhythmic breathing.	Transvenous stimulation of phrenic nerve during apneas.	Multicenter, randomized, open-label study *n* = 151	Reduction in AHI and freedom from serious adverse events	The remedē System Pivotal Trial showed significant reduction in AHI, arousal index, desaturation and apnea episodes. It also revealed improvement in quality of life, sleep structure and EF [10,11]	Cumulatively, 21 (14%) serious adverse events were observed in 5-year follow-ups (15; (10%) in the first 12 months). It predominantly included electrode dysfunction, electrode dislocation and infection of the implantation site [10]
Asymptomatic diaphragmatic stimulation	High left ventricle pre-load and after-load pressures increase remodeling and HF progression.	Stimulation of diaphragm muscle fibers synchronized with cardiac cycle to decrease intrathoracic pressures.	Single-center, randomized, open-label study *n* = 33	LVEF improvement	EPIPHRENIC II Study showed significant improvement of LVEF, maximal power on effort, reduction in NYHA class, without differences in 6-min walking test or BNP concentration [12,13]	Three patients were excluded due to dysfunctional diaphragmatic electrode. No adverse events were observed [12]
			Multicenter, non-randomized, open-label study *n* = 15	Freedom from serious adverse events during procedural recovery or acute therapy	VisONE study showed improvement in LVEF and life quality (evaluated in SF-36); extended walking distance during the 6 MWT was observed at a 1-year follow-up. [13]	No adverse events were observed during procedural recovery, acute therapy (primary outcome) and in 12month follow-up (secondary outcome) [13]
Reprieve system	Problems with controlling decongestive therapy to avoid too rapid diuretic response and hypovolemia and, on the other hand, providing too much fluid, which worsens volume overload.	Sustaining the accurate fluid balance by measuring the urine output and providing the exact amount of replacement solution to achieve preset fluid balance.	Non-randomized, single-center, prospective, open-label, studies, both *n* = 19	Device and procedure-related adverse events and decongestive efficacy	Higher urine output and decrease in CVP in comparison to the baseline. Actual fluid loss did not exceed target fluid loss at the end of therapy in every patient [14]	No serious adverse events were observed. One case of hypokalemia occurred.
Transcatheter renal venous decongestion system	Congestion in renal veins.	Transfemoral inserted flow pump, which reduces renal vein pressure to the desired level.	No results have been published so far.	Device and procedure-related adverse events, technical and procedural feasibility	The trial to evaluate TRVD was terminated prematurely, no results have been published so far.	No results have been published so far.
Doraya Catheter	Congestion in renal veins.	Partial obstruction of the flow in the inferior vena cava below the level of the renal veins reduces renal vein pressure	First in-human, single-arm, open-label study *n* = 9	Serious adverse events.	The catheter was successfully deployed in all patients. Clinical symptoms, as well as diuresis and natriuresis, improved [15]	No device-related or embolic events were reported. One serious procedure-related adverse event: bleeding hematoma from the injection site, resolved without sequelae.
preCARDIA	Increased right ventricle preload.	Obstruction of the superior vena cava leading to an intermittent decrease in preload.	Multicenter, prospective, single-arm exploratory safety and feasibility, open-label, trial *n* = 30	Freedom from device or procedure-related serious adverse events	Successful decrease in right atrial pressure and PCWP, increase in net fluid balance and urine output [16]	No device or procedure-related serious adverse events were observed.
WhiteSwell	Increased preload causes lymphatic congestion, which impairs interstitial drainage and exacerbates oedema.	Reduction in the pressure in the area of lymphatic duct outflow into venous vessels.	The animal model study, *n* = 7 sheep, used in 1 human, *n* = 1	Serious adverse events.	Examined in a ovine model. Trend toward improved oxygenation an diuresis was noticed [17]	No adverse events were reported in in-human application.
AquaPass	Insufficient urine volume removal.	Enhancing the sweat rate to remove fluid directly from interstitial space.	Feasibility and short-term performance, single-arm, open-label study, *n* = 16	Serious adverse events, treatment tolerance, ability to control skin temperature between 33 and 38 Celsius degrees).	The procedure was safe in HF patients, successful weight loss was observed. Increased skin temperature without elevating core temperature above average was achieved in each patients [18]	No adverse event occurred.

Abbreviations: CVP—Central Venous Pressure, SNS—Sympathetic Nervous System, RAAS—Renin-Angiotensin-Aldosterone System, HF—Heart Failure, MANCE—major adverse neurological or cardiovascular system or procedure-related event rate, MLWHF QOL—Minnesota Living With Heart Failure Quality of Life, NT-proBNP—N-terminal pro brain natriuretic peptide, EF—ejection fraction, PNS—parasympathetic nervous system, NYHA—New York Heart Association, PAMP—Pulmonary Arterial Mean Pressure, PCPW—Pulmonary Capillary Wedge Pressure, HFpEF—Heart Failure with Preserved Ejection Fraction, LV—left ventricle, AHI—Apnea-Hypopnea Index, LVESV—Left ventricle end-systolic volume, LVESD—Left ventricle end-systolic dimension, TRVD—transcatheter renal venous decongestion system, 6 MHW—Six Minute Hall Walk Test.

## Data Availability

Not applicable.

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
