# Peer review of "Novel Therapeutic Devices in Heart Failure"

_jcm, 2022, doi:10.3390/jcm11154303_

Round 1

Reviewer 1 Report

This is a nicely written review of novel therapeutic devices in heart failure. Guzik et al focus on new therapeutic targets and devices. The authors should be congratulated for their effort.  I only suggest, if possible, including a central figure that may summarize the main aspects of the review. Considering that the article is relatively long, I think that it may increase the interest of the readers.

Author Response

Dear Sir or Madam,

We are grateful for your review and appreciate your suggestion.

We enriched our manuscript with a general illustration.
We believe that the manuscript has been further improved and would match your expectations.

Reviewer 2 Report

I read with interest the paper by Guzik et al. titled ”Novel therapeutic devices in heart failure”.

I thank the authors and the editor for the opportunity to review this manuscript.

However, I have some reservations as detailed below.

Major points

It should be made clear that it is a literature review and not a systematic review and this should be noted in the Limitations.

A review was recently published on this topic (J Am Coll Cardiol:2021:78:931-956) which cover many of the novel therapies presented in the manuscript in greater detail.

A lot of the writing is unclear and important parts of the included studies are not presented.

The authors should briefly describe the available data on adverse outcomes for each of the novel therapies presented in the manuscript and comment on the severity of adverse outcomes where applicable.

Where relevant, please make sure to describe whether the studies demonstrated any effect on hard outcomes such as heart failure hospitalizations and mortality or whether such data is not available.

Please also note where applicable whether studies were large or small and further, it should be noted when studies were not RCTs.

There are many grammatical and writing issues. I have commented on some of them below. Consider to use a spell checker, use e.g. Grammarly or similar.

See specific points below.

Abstract

”improve the outcome” – ”improve clinical outcome”.

 1. Introduction

“indirect targeting HF biological pathways” - “indirect targeting of HF pathways” or “indirect targeting of biological HF pathways”.

The authors should consider to include a brief description of heart failure, e.g. reduced LVEF, elevated filling pressures, and clinical symptoms.

”e.g. methods to manipulate sympathetic”

“especially those that are permanently implanted”.

L31-33 – please revise for clarity and grammar.

”insightful assessment in a large” – unclear, try e.g. ”thorough assessment in large”.

” well-organized space” – please revise the wording.

”adrenergic receptors (AR)” – there are many abbreviations in the manuscript, consider to avoid this one.

“HF is the state with oversecretion of catecholamines that continuously stimulate cardiovascular AR.” – please revise the wording or omit this sentence as it is a repetition of the previous.

L55-58 – For clarity this might be moved to the first part of this paragraph.

”has been already reflected ” – ”is reflected”.

2.1. Baroreflex activation therapy

”patients quality of life” – “quality of life”.

”possibly reduces hospitalisations in patients” – It should be made clear that there was only a trend (P = 0.08) or better this should be omitted as it was only a trend for a secondary outcome.

“Dell’Oro et al. demonstrated that BAT significantly improved EF” – Only 7 patients were included in this follow up study whereby the wording of the description of the results should be softened.

In the discussion of the results of the different studies, it may be noted what the primary outcome of each study was. Many of the positive effects of the different therapies seem to be secondary outcomes.

2.2. Vagus nerve stimulation

“and physical component” – unclear.

There was no control group in the ANTHEM-HF trial. This should be noted when the evidence is presented, please revise throughout.

“VNS, further multiethnic investigations are needed.” – Do the authors mean “multicentric”?

2.3. Splanchnic nerve modulation

“which prevents the exercise provoked a rapid shift of blood” – unclear, please revise.

”that underwent SNM procedure” – consider to add which procedure, surgical? Please revise the sentence for grammar and clarity.

L149-150 the exact numbers seem  be less relevant here. At least the decimals can be omitted.

”treatment of extra cardiovascular diseases” – consider to omit or replace by “non-cardiac diseases”.

2.4. Cardiac pulmonary nerve stimulation

“(cardiac muscle relaxation – diastolic function)” – Do the authors mean “improved diastolic function”? Please revise also for systolic function.

3. Respiratory disturbances in heart failure

“cause the intensification of pathophysiological pathways convergent” – please revise for clarity and concision.

”adaptive servo-ventilation (ASV)” – there are a great number of abbreviations, this and others can be omitted.

“This results in enhancing the pathways at the cellular and neurohormonal levels described above.” – it might be more clear and short to just state that this increases SNS and RAAS activity.

3.1. Phrenic nerve stimulation

“It also anticipates the intensification of pathogenetic pathways characteristic of sleep apnea” – please revise for clarity.

“phrenic nerve stimulation, a significant reduction” – please add a “found” or “demonstrated” or similar.

”and improved sleepiness”.

Delete “were observed”.

”5-years observation” – should this be ”at 5-years follow up”?

”have experienced” – ”had an”.

L236-8: Please add similar safety data for the other device therapies.

3.2. Synchronized diaphragmatic therapy

L246 Please revise “The element” to e.g. “An element” or “A factor”

“of the type I diaphragm” and “causing its tension to change the pressure in the chest“ – please revise for clarify.

“it does not cause effective breathing movement” – unclear.

“in a one-year follow-up” – “at one-year follow-up”.

“the promising results of the trial” – trials?

4. Novel techniques to facilitate decongestion

L275 – please revise for grammar. Try to delete “As the” and combine the sentence with the following.

”challenge for the treatment” – “clinical challenge”.

4.1. Reprive therapy®

L283-5: Unclear, please revise all parts of the sentence for clarity and grammar.

“paradoxically to so-called diuretic resistance” – why would it be paradoxical that diuretics don’t work in a dehydrated patient?

”The trend” – ”A trend”. There is so much material to cover in the manuscript that it seems superfluous to mention trends in secondary outcomes.

”has been noticed” – ”was observed”.

“one, small, single-centre, one-arm study” – unclear, above it seems that there were two studies with more than one arm?

It might also be noted that it seems to be intended for acute HF.

“have been already under clinical investigation.” – do the authors mean “are under clinical investigation”?

4.2. Transcatheter renal venous decongestion system (TRVD) and Doraya Catheter

”As renal veins” – ”As renal vein”.

“Test in the porcine model” – “The device was tested in a porcine model”.

”the suprarenal balloon” – “a suprarenal balloon”. Please revise the articles carefully throughout the manuscript.

”Doraya Catheter” – ”The Doraya Catheter”.

“to perform a transient modification of renal venous pressure” – “to temporarily reduce renal venous pressure”.

323-9: Please revise for concision.

”, have improved” – ” all improved”.

4.3. preCARDIA

“In that case, the device is implemented to” – “The device is inserted in the”.

”After the 24 hours of the treatment, 130%” – “At 24 hours of treatment an 130%”.

It seems that fluid balance increased (higher weight?) as urine output increased? Please revise.

“in terms of thrombotic events, strokes” – was the risk high?

“Performed studies have the first-in-human design” – unclear.

4.4. WhiteSwell®

”the sheep” – ”a sheep”.

374-6: please revise for clarity and concision.

“study character of the performed investigation” – delete.

”the wider” – ”wider”.

4.5. AquaPass

“The Aquapass is the system of enhancing” – try e.g. “The Aquapass system enhances sweat rate and thus fluid removal”.

“not to cause the awareness of the perspiration” – unclear, consider to omit.

Is the apparatus applied for a short time or meant to be used all day?

”included a limited number of” – ”included only 6 healthy subjects and 10 HF patients”.

”successfully increased” – ”increased”.

Table 1

Please add a column on adverse effects and a column on trial designs and size.

“NECTAR-HF presented significant improvement“ – “NECTAR-HF presented significant improvements”.

”partial pressure, which is below“ –“partial pressure to below”.

“Synchronized with heart cycle stimulation of diaphragm muscle fibres type I to change intrathoracic pressure” – try e.g. “Stimulation of diaphragm muscle fibres synchronized with the cardiac cycle in order to ?lower? intrathoracic pressure”

”Improve of EF” – “Improvement in LVEF”.

“and a trend toward increased natriuresis” – consider to omit trends.

”have improved[46].” – “improved[46].”

Increeas – typo.

”Examined in the porcine model“ – “Examined in a porcine model”.

“Procedure was safe in HF” – “The procedure was safe in HF”.

5. Conclusions

“which have not been used in therapy until recently” – Please revise, the novel therapies are not yet used in clinical practice.

References

The doi of published articles are unnecessary. Please use abbreviated journal titles according to journal policy.

I have no conflicts of interest to declare.

Round 2

Reviewer 2 Report

I commend the authors for the revision, the manuscript is improved

I have some remaining comments.

There are still some issues with the writing and grammar and I have commented on some of them below.

”Every instance has been” – ”Each segment is”.

”the elevation of” – ”elevated”.

“are essential for the episode of” – delete.

”to improve the 36 HF outcome” - ”to improve outcome in HF”

“before we decide to use them” – “before they can be used”.

”Disbalance” – Imbalance.

Disbalance of the autonomic nervous system plays a crucial role in the pathogenesis of HF. The HF syndrome is undoubtedly a state with ANS dysregulation, in which activity of the sympathetic nervous system (SNS) exceeds the buffer capabilities of the parasympathetic nervous system (PNS). –  can be shortened to e.g.  “Imbalance of the autonomic nervous system plays a crucial role in the pathogenesis of HF as the sympathetic nervous system (SNS) exceeds the buffer capabilities of the parasympathetic nervous system (PNS).”

”As ANS” – ”The ANS”. Replace ” thus” by ”whereby”.

“ANS recep-61 tors are highly specialized sensors, responsible for creating adequate impulses in response 62 to stimuli.” – consider to delete.

“is an area for searching for HF therapies” – try “is an area for ongoing research in HF therapies”.

”reacts in increasing” – ”reacts by increasing”.

”Several clinical research” – ”Several clinical studies have”.

“patients (264 participants), meeting the FDA-approved instructions for BAT enrolment criteria for BAT” – “264 patients with”.

”EF 35” – a symbol is missing.

”resistant arterial hypertension” – ”drug resistant arterial hypertension”.

”BAT is incorporated” – ” BAT can be incorporated”.

“,a critical therapeutic target in HF” – repetition.

“VNS in heart failure with reduced ejection fraction HFrEF" - VNS in HFrEF”.

“showed statistically significant improvement” – can be shortened to “showed improvements”. Please revise throughout the manuscript.

“without positive changes in echocardiography” – “without changes in echocardiographic measures”.

“NEural Cardiac TherApy foR Heart Failure (NECTAR-HF” – “The NEural Cardiac TherApy foR Heart Failure trial (NECTAR-HF”. Please revise similar instances throughout the manuscript.

”60 participants) uncontrolled” – ”60 participants, uncontrolled design)”.

”upper abdomen viscera” - ”upper abdominal viscera”  or ”upper abdominal organs”.

”whereby protects against” – “and thereby prevents”

”increased physical exercise” – “physical exercise”.

“to whom the same procedure was applied” – “who underwent the same procedure”.

“quality of life in treated patients after 12 months of the procedure” – “quality of life at 12 months after the procedure”.

”1 month after SNM procedure” – ” 1 month after the procedure”.

“electrode through pulmonary arteries” – should it be “electrode placed in a pulmonary artery”?

”stimulates surrounding” – ”stimulates the surrounding”.

“affecting both these systems: cardiovascular and respiratory” - “affecting both the cardiovascular and respiratory system”.

L235 please use the abbreviation SNS which is already defined, i.e. “SNS and RASS activation”.

Please revise the abbreviations throughout the manuscript.

“These mechanisms are 237 common to the OSA/CSA and HF pathophysiology. Their intensification causes acceler-238 ation of HF progression” – “These mechanisms are common to the OSA/CSA and HF pathophysiology and accelerates HF progression”.

L249-54: seems irrelevant for the present discussion.

L258-66 this introduction to the pathophysiology should be moved and incorporated into L228-31 for clarity and concision.

”relatively stable” - ”maintain a relatively stable pO2 and pCO2 and prevent over-activation of SNS and RAAS”.

”reduction of sleepiness” – ”reduced sleepiness”.

Table 1: Phrenic nerve stimulation  “life quality” – “quality of life”.

”the 295 severe-adverse-effects”- “a severe-adverse-effect”.

“The assessment of phrenic nerve stimulation impact on mortality in HF patients with 300 CSA syndrome as well as a large scale clinical trial is needed.” – try “The effect of phrenic nerve stimulation on mortality in HF patients with CSA syndrome is unknown and large scale clinical trials are needed.

“A factort that causes chronic stress on the heart muscle with increased in tracardiac pressures and the progression of HF is elevated intrathoracic pressure” – unclear and long. Try something like: “”Elevation of intrathoracic pressure causes chronic stress on the heart muscle and may worsen HF. “ Consider to start by stating why intrathoracic pressure is increased.

“aims to synchronise with heart work cycle stimulation of diaphragm’s” – I do not entirely understand this sentence maybe, try: “aims to synchronize the cardiac work cycle to changes in diaphragm movement by stimulation of …”

“which in turn allows the intra-thoracic pressure change” – “which in turn reduces intra-thoracic pressure” ?

”The minimally invasive method” – ”A minimally invasive method”.

“in procedural recovery or acute therapy and during 12 month” –unclear, can this be shortened to “at 12 month”

“It was shown that nearly 50% of the classi-342 cally treated HF patients are discharged with the residual congestion” – “Up to nearly 50% of the classically treated HF patients are discharged with residual congestion”.

“a need for novel fluid removal techniques has emerged” – be careful about the “unmet needs” arguments when there is not enough data support a treatment. Try: “interest in novel fluid removal techniques has emerged”.

“for the more accurately controlled (and therefore may 350 be safer) decongestion for HF patients” – unclear, please revise.

L 349: If placement of a urinary catheter and vein cannula is required, the technique is not non-invasive.

Table 1 Reprive system : “studies n=19” should be “studies, both n=19”

”No serious adverse effects, hypokalemia” – please revise.

”showed the safety and efficacy” – unclear, what did the study show?

“are under clinical research” – “are being investigated”.

Table 1 Doraya Catheter “in inferior vena cava below the renal veins level reduces the pressure in renal veins” - in the inferior vena cava below the level of the renal veins level reduces renal vein pressure”.

“First in-human study n=9 ” – please revise throughout the table for relevant information such as “single arm”, non-controlled”, “randomized”, “open label” etc. were applicable.

“– 1 procedure related ad-verse effect occurred” delete as this is stated in the column on the right. Please revise throughout the “objective” column, it should state the outcome, not the result.

”Catheter was successfully deployed. ” – ”The catheter was successfully deployed in all patients.”.

 “No device-related or em-bolic events. “ – “ No device-related or em-bolic events were reported”.

”By causing the partial block of the flow, ” – “By partially blocking venous flow, the ”.

”resolved 410 without sequelae”- “that resolved 410 without sequelae”.

”on limited population” – ”in a small population”.

”Performed studies were” – “The studies are”.

“its impact on the out-437 come”- “its impact on out-437 come”.

”it plays a crucial role” – “it could play a role” or revise by stating that it plays a role in the homeostatic process of fluid removal from tissues and delete the following sentence.

”the device designed” –”a device designed”.

Table 1 WhiteSwell: ”Increase in CVP” – try “Increased preload”

”Examined in a porcine model” – should be ”ovine”

“No technical issues were observed” – This is contradicted by L469.

“A study (NCT04578353) included included only 6 healthy subjects and 10 HF patients” – “The AquaPass system was evaluated in a study (NCT04578353) including only 6 healthy subjects and 10 HF patients who underwent three treatment sessions for up to 4 hours”.

Please revise table 1 with an aim to make the presentation of outcomes, results and adverse effects more uniform across the studies.

“Importantly, our review is a literature review” – “Importantly this is a literature review.
